# Differences in the subgingival microbiome according to stage of periodontitis: A comparison of two geographic regions

**Gloria Inés Lafaurie**[1]*, **Yineth Neuta**[1], **Rafael Ríos**[2], **Mauricio Pacheco-Montealegre**[2], **Roquelina Pianeta**[3,4], **Diana Marcela Castillo**[1], **David Herrera**[3], **Jinnethe Reyes**[2], **Lorena Diaz**[2], **Yormaris Castillo**[1], **Mariano Sanz**[3], **Margarita Iniesta**[3]

1 Unit of Basic Oral Investigation -UIBO, School of Dentistry, Universidad El Bosque, Bogotá, Colombia, 2 Molecular Genetics and Antimicrobial Resistance Unit, Universidad El Bosque, Bogotá, Colombia, 3 ETEP (Etiology and Therapy of Periodontal and Peri-Implant Diseases) Research Group, School of Dentistry, University Complutense of Madrid (UCM), Madrid, Spain, 4 School of Dentistry, Corporación Universitaria Rafael Núñez, Cartagena, Colombia

* lafauriegloria@unbosque.edu.co

**Data Availability Statement:** Original data of samples evaluated are available on NCBI

## Abstract

No microbiological criteria were included in the 2018 EFP-AAP classification of periodontal diseases that could be used to differentiate between stages and grades. Furthermore, differences in the subgingival microbiome depending on stage and grade have not been established. Sixty subgingival biofilm samples were collected in Spain (n = 30) and Colombia (n = 30) from three distinct patient categories: those with periodontal health/gingivitis (n = 20), those with stage I-II periodontitis (n = 20), and those with stage III-IV periodontitis (n = 20). Patients were evaluated by 16S rRNA gene amplification sequencing. Amplicon sequence variants were used to assign taxonomic categories compared to the Human Oral Microbiome Database (threshold ≥97% identity). Alpha diversity was established by Shannon and Simpson indices, and principal coordinate analysis, ANOSIM, and PERMANOVA of the UNIFRAC distances were performed using QIIME2. Although differences in the alpha diversity were observed between samples according to country, *Filifactor alocis*, *Peptostreptococcaceae* [XI][G-4] *bacterium* HMT 369, *Fretibacterium fastidiosum*, *Lachnospiraceae* [G-8] *bacterium* HMT 500, *Peptostreptococcaceae* [XI][G-5] *[Eubacterium] saphenum*, *Peptostreptococcus stomatis*, and *Tannerella forsythia* were associated with periodontitis sites in all stages. However, only *F. alocis*, *Peptostreptococcaceae* [XI][G-4] *bacterium* HMT 369, *Peptostreptococcaceae* [XI][G-9] *[Eubacterium] brachy*, *Peptostreptococcaceae* [XI][G-5] *[Eubacterium] saphenum*, and *Desulfobulbus* sp. HMT 041 were consistent in stage III-IV periodontitis in both countries. *Porphyromonas gingivalis* and *Tannerella forsythia* were differentially expressed in severe lesions in the countries studied. Although some non-cultivable microorganisms showed differential patterns between the different stages of periodontitis, they were not the same in the two countries evaluated. Further studies using larger samples with advanced next-generation techniques for high-throughput sequencing of phyla and non-cultivable bacteria within the subgingival microbiome could provide more insight into the differences between stages of periodontitis.

(BioProject PRJNA828047). All code used to generate statistics and figures as well as raw data are available on: https://www.ncbi.nlm.nih.gov/bioproject/828047.

**Funding:** Funding: This study was funding to Ministerio de Ciencia, Tecnología e Innovación-MINCIENCIAS, Colombia (grant number 130880763942), and Cátedra Extraordinaria Dentaid de Investigación en Periodoncia (UCM) in Spain. The funders had no role in study design, data collection and analysis, decision to publish, or preparation of the manuscript.

**Competing interests:** The authors have declared that no competing interests exist.

## Introduction

Periodontitis is a multifactorial chronic inflammatory disease associated with dysbiotic changes within the subgingival biofilm, leading to hyperinflammatory and immune reactions that destroy the supporting tissues of the teeth [1]. Although the incidence of periodontitis may vary among populations, depending on economic, cultural, social, and ethnic factors [2,3], the prevalence of severe periodontitis has remained relatively stable in the last few decades and is estimated to be approximately 10% in the adult population [4,5]. However, mild to moderate periodontitis has been associated with wide variations in prevalence when comparing different ethnicities and geographical environments, with the prevalence ranging from 12% to 55% [5]. Similarly, cross-sectional association studies in other parts of the world have reported significant variations in the prevalence of predominant target bacteria within oral/subgingival biofilm [6,7].

The Human Oral Microbiome Database (HOMD) has reported at least 700 bacterial taxa at the species level [8,9]. Furthermore, the current use of next-generation sequencing (NGS) has allowed the study of new non-culturable or difficult to culture microbial genera, which may also be strongly associated with diseases such as periodontitis and could be used as diagnostic and therapeutic targets as classical periodontal pathogens derived from culture-based microbiological studies have been used [10]. The microbiome has been studied using various approaches. Using DNA homology, specifically the 16S ribosomal RNA gene, several studies have evaluated microbiome richness, diversity, and organization in taxonomic groups [11]. Recent studies have analyzed microbial diversity using amplicon sequence variants (ASVs), which, through unique gene sequences rather than consensus, reduce classification bias because each ASV is taxonomically assigned independently rather than being grouped into an organizational taxonomic unit (OTU) [12].

Studies that evaluated the association between the microbiome and subgingival biofilm reported differences in the microbial composition when comparing individuals with periodontitis among different populations [13–15]. However, it is unclear whether these differences are due to the analysis of microbiome data or whether there are distinct differences in the microbiome depending on the clinical diagnostic criteria or in the different populations studied [6,7].

The recent classification of periodontitis in 2018 incorporated clear criteria of disease severity, complexity, and patterns of progression to define the different stages and grades currently used [1]. However, this classification did not incorporate any etiological criteria to differentiate among these stages or grades due to the lack of evidence-based scientific data on the use of microbiological and host response diagnostics in patients with periodontitis. Therefore, this study aimed to use NGS technologies to examine the subgingival microbiome of patients with different periodontal diagnosis based on the current classification criteria [16]. Furthermore, although the current microbial signatures may be capable of discriminating between ethnicities in saliva samples from different populations [6,7], the presence of unculturable and difficult to culture bacteria in subgingival biofilms using DNA sequencing strategies when comparing different periodontitis populations has not yet been reported. Therefore, this cross-sectional observational study also aimed to compare the composition and diversity of the subgingival microbiome in patients with different periodontal stages from two distinct geographical populations (Spain and Colombia).

## Materials and methods

### Study sample

This cross-sectional observational study was conducted once the respective clinical ethics committees approved the study protocol (approval 012–2018 in Colombia and 18/127-E in Spain).

This study adhered to the international ethical guidelines of the Declaration of Helsinki for human experimentation.

Sixty individuals were matched for periodontal condition (stage), sex, and age (difference not more than 5 years) and allocated into three groups: (1) periodontal health/gingivitis (n = 20); (2) stage I-II periodontitis (n = 20); (3) stage III-IV periodontitis (n = 20), with ten individuals from Spain and Colombia in each group. After collecting the sociodemographic (age, sex, country) and medical data (by answering a medical questionnaire), the participants were asked to sign an informed consent form if they were willing to participate and fulfilled the below criteria.

**Inclusion criteria.** Individuals between 30 and 65 years of age with a periodontal diagnosis according to the 2018 classification as follows were included [16]:

1. Periodontal health or gingivitis with no clinical attachment loss (CAL), radiographic bone loss (RBL), and pocket depth (PD) $\leq$ 3 mm.

2. Stage I or II periodontitis with interdental CAL of 1–2 mm (stage I) or 3–4 mm (stage II) and RBL affecting only the coronal third of the root.

3. Stages III or IV periodontitis with interdental CAL > 5 mm, and RBL extended to the middle or apical third of the root.

Indirect estimation of the pattern of disease progression was performed considering bone loss as a function of age at the most affected tooth (RBL expressed as a percentage of root length divided by the age of the patient), and patients were categorized as follows: a) grade A, percentage of RBL $\div$ age <0.25; b) grade B, percentage of RBL $\div$ age 0.25 to 1.0; and c) grade C, percentage of RBL $\div$ age> 1.0.

**Exclusion criteria.** The exclusion criteria were: 1) previous non-surgical and surgical periodontal treatment within the last year; 2) acute periodontal conditions such as periodontal abscesses or necrotizing periodontal diseases at the time of evaluation; 3) antibiotics in the last three months; 4) systemic diseases or conditions (e.g., diabetes, immune system disorders) that may influence the periodontal condition; 5) pregnant women; and 6) use of anti-inflammatory drugs, anticonvulsants, immunosuppressants, and calcium channel blockers at the time of sample collection or six months before the study.

## Clinical examination

Two calibrated examiners evaluated the clinical parameters (one from each country) using a UNC-15 periodontal probe (HuFriedy, Leimen, Germany) at six sites per tooth in all teeth, except for the third molars and dental implants. The clinical parameters were PD and CAL, expressed in millimeters (mm); plaque index (PlI); and bleeding on probing (BoP), expressed as the percentage of positive sites in the mouth. Intra-examiner calibration was performed by recording duplicate PD and CAL measurements in three patients twice during the same visit, at 30-min intervals. The interclass correlation coefficient showed 90.2% agreement for PD and 89% for CAL in Colombia, and 86.3% and 84.7% in Spain, respectively.

## Microbiological sampling

After isolation from the saliva and removal of the supragingival biofilm from the sampling area, samples were taken from four sites by placing two consecutively inserted sterile paper points per site, which were left at the bottom of the sulcus/pocket in place for 10 s. The sampling sites were selected as follows: in periodontal health or gingivitis, samples were taken from the mesiobuccal sites of the first molars and, when absent, from the adjacent second

molars. Subgingival samples were collected from the most accessible site with the deepest PD and BoP in each quadrant in individuals with periodontitis.

## DNA extraction

DNA was extracted from all samples using an extraction kit (QIAamp Mini Extraction Kit) following the manufacturer's recommendations. Once the DNA was obtained, the DNA quality was verified by determining the absorbance radii 260/280 and 260/230 for purity verification on a NanoDrop™ (NanoDrop™ 2000). We used a fluorometer (Life Technologies, Invitrogen) to quantify double-stranded DNA. Once the amount was higher than 1.0 μg/mL and purity was confirmed, we conducted 16S rRNA gene amplification, sequencing, and processing.

## 16S rRNA gene amplification, sequencing, and processing

To identify the microbial diversity in the collected samples, high-throughput sequencing of V3-V4 16S rRNA (primers 515F and 806R) [17] was used in all the samples, DNA was obtained using the extraction kit (QIAamp mini extraction kit), and library preparation was performed in a two-step polymerase chain reaction (PCR) procedure [18]. The PCR products from both amplifications were analyzed on agarose gels (1.5%), and DNA concentrations were quantified using a Qubit™ dsDNA HS and BR Assay Kit. Samples were pooled to equimolar concentrations and pair-end sequenced (250 nt reads) on an Illumina MiSeq machine. ASVs were detected using the DADA2 plugin in QIIME2 version 2020 [19]. The process included the quality trimming of reads, elimination of replicates, and chimera filtering. ASVs with fewer than ten sequences were discarded. ASVs were assigned to taxonomic categories using NCBI BLASTn version 2.2.2320 against the expanded HOMD (eHOMD) [20]. An identity threshold of ≥97% was used to assign ASVs to taxonomic categories.

## Statistical analyses

The Shapiro-Wilk test was used to evaluate the distribution of continuous data. Absolute and relative frequencies were used to estimate the categorical variables. Clinical and sociodemographic data were calculated by periodontal status group and country and compared using an analysis of variance, Bonferroni test, Kruskal–Wallis test, Mann–Whitney U test, chi-square, or Fisher tests with a significance level of 5% ($p \leq 0.05$).

Alpha diversity metrics, including the richness (observed taxa), Shannon index, and Simpson index, were calculated using Past version 4.07b22 [21]. The Bray–Curtis similarity score was obtained for each sample against each other [22], and weighted and unweighted UNIFRAC distances were obtained using QIIME2 [19]. Principal coordinate analysis, ANOSIM, and PERMANOVA of the UNIFRAC distances were performed using QIIME2 [19].

Alpha diversity metrics were compared using the Kruskal–Wallis/Mann–Whitney U tests to test for differences among the diagnostic subject categories. We performed Welch's t-test [23] using STAMP version 2.1.3 software to identify possible differences in specific taxa between subject groups and countries (Colombia and Spain), smokers and non-smokers, and A/B grades vs. C grades.

# Results

## Clinical and sociodemographic characteristics of the sample studied

The selected participants were well matched for sociodemographic and clinical variables and current smokers when compared by country in the three groups (p>0.05). However, in Colombia, PII was higher in participants with stage I-II periodontitis than in healthy

participants with gingivitis (p = 0.02). In Spain, participants with stage III-IV periodontitis demonstrated higher BoP than participants with periodontal health/gingivitis (p = 0.05) (Table 1).

## Subgingival microbiome composition and diversity in the complete sample and by stage of periodontitis

The subgingival microbiome was analyzed by sequencing amplicons in the hypervariable (V3-V4) region of the 16S-rRNA gene for both composition and diversity. A total of 85,864 sequencing reads were analyzed, and 2704 ASVs were obtained, of which 2447 were assigned to taxonomic categories. However, 257 reads had less than 97% identity with the eHOMD [20].

Across all the samples, 295 taxonomic categories were identified, with the most abundant phyla being *Firmicutes* (39%), *Fusobacteria* (21.2%), *Bacteroidetes* (12%), and *Proteobacteria* (11.9%). In samples from individuals with periodontal health/gingivitis, the most abundant phyla were *Firmicutes* (35.9%), *Fusobacteria* (21.9%), and *Proteobacteria* (16.9%). In contrast, in samples from periodontitis in stage I-II and III-IV patients, the most abundant phyla were *Firmicutes* (38.5% and 43.1%, respectively), *Fusobacteria* (22.1% and 19.2%, respectively), and *Bacteroidetes* (15.1% and 14.6%, respectively) (Fig 1A).

The most abundant species across all patients analyzed (>1% of the total count) was *Fusobacterium* sp. HMT 203 (7.6%), *Fusobacterium nucleatum* subsp. *vincentii* (7.4%), *Streptococcus mitis* (5.4%), and *F. alocis* (2.7%). In individuals with periodontal health/gingivitis, the most abundant species were *S. mitis* (8.3%), *F. nucleatum* subsp. *vincentii* (6.5%), and *Fusobacterium* sp. HMT 203 (5.3%). In patients with stage I-II periodontitis, the most abundant species were *Fusobacterium* sp. HMT 203 (9.1%), *F. nucleatum* subsp. *vincentii* (6.6%) and *S. mitis* (4.1%), and in patients with stage III-IV periodontitis, the most abundant species were *F. nucleatum* subsp. *vincentii* (9.1%), *Fusobacterium* sp. HMT 203 (8.3%), and *F. alocis* (4.6%) (Fig 1A).

There were differences in the alpha diversity between participants with periodontal health/gingivitis and stage I-II and III-IV periodontitis, with the three indices using the entire sample. Based on Shannon's and Simpson's indices, periodontal health/gingivitis samples had lower richness, with a more even distribution of bacterial communities (Fig 1B). Similar results were obtained with the PERMANOVA and ANOSIM tests when assessing the weighted and unweighted UNIFRAC distances [19], with significantly different patterns seen when comparing periodontal health/gingivitis with stage I-II and III-IV periodontitis samples (p≤0.01 in all comparisons). However, there were no statistically significant differences when comparing samples from stage I-II and III-IV periodontitis across all the different metrics used, either in smokers or non-smokers (data not shown).

Principal coordinate analysis of weighted and unweighted UNIFRAC distances (48% and 31% explained variance, respectively) showed that the microbial patterns of periodontal health/gingivitis samples clustered away from those of periodontitis (Fig 2). However, microbial patterns from Colombia's periodontal health and gingivitis sites formed a small cluster with some periodontitis samples, indicating similarities in their microbial diversity. This result was corroborated using the Bray–Curtis similarity index (Figs 2 and 3).

When comparing the proportions of different species between periodontitis and health/gingivitis samples in the entire sample, *F. alocis* and *Peptostreptococcaceae* [XI][G-4] *bacterium* HMT 369, *F. fastidiosum*, *Lachnospiraceae* [G-8] *bacterium* HMT 500, *Peptostreptococcaceae* [XI][G-5] *[Eubacterium] saphenum*, *Peptostreptococcus stomatis*, and *T. forsythia* were significantly associated with periodontitis in both stages of periodontitis ($p \leq 0.01$) (Table 2).

**Table 1. Sociodemographic characteristics by country and periodontal status.**

| | PERIODONTAL STATUS | | | Differenceamong groups | Differencebetween countries |
|---|---|---|---|---|---|
| | Health/Gingivitis | Periodontitis I-II | Periodontitis III-IV | *p-value* | *p-value* |
| **Country (n)** | | | | | |
| Colombia | 10 | 10 | 10 | NS | NS |
| Spain | 10 | 10 | 10 | | |
| **Age [mean (SD)]** | | | | | |
| Colombia | 44.7 (12) | 40 (9) | 42.3 (7) | NS | NS |
| Spain | 41.1 (9.6) | 45 (6.9) | 43.8 (7.8) | | |
| **Gender** | | | | | |
| **Female/Male [n (%)]** | | | | NS | NS |
| Colombia | 7(23)/3(10) | 6(20)/4(13) | 5(17)/5 (17) | | |
| Spain | 7(23)/3(10) | 6 (20)/4(13) | 5(17)/5 (17) | | |
| **Smokers [n (%)]** | | | | NS | |
| Colombia | 2 (20) | 1 (10) | 2 (20) | | NS |
| Spain | 1 (10) | 2 (20) | 3 (30) | | |
| **No smokers [n (%)]** | | | | NS | NS |
| Colombia | 8 (80) | 9 (90) | 8 (80) | | |
| Spain | 9 (90) | 8 (80) | 7 (70) | | |
| **Plaque index (PlI)** **Median (IQR)** | | | | ¥0.010 | |
| Colombia | 19 (12–33) | 98 (25–100) | 66 (60–100) | [a]**0.030** | [a]**0.02** |
| Spain | 53 (13–38) | 78 (4–85) | 79 (48–100) | [b]**0.005** | [b]NS |
| | | | | [c]NS | [c]NS |
| **Bleeding on probing (BoP)** **Median (IQR)** | | | | ¥0.0001 | |
| Colombia | 8 (4–13) | 100 (46–100) | 66 (1–60) | [a]**0.0007** | [a]NS |
| Spain | 16 (2–27) | 45(19–55) | 79 (1–98) | [b]**0.0001** | [b]**0.05** |
| | | | | [c]**0.0291** | [c]NS |
| **Pocket depth (PD)** **Median (IQR)** | | | | ¥0.0001 | |
| Colombia | 2.35 (2.13–2.65) | 3.22 (2.9–4) | 3.83 (3.4–4.7) | [a]**0.0001** | [a]NS |
| Spain | 2.32 (2.1–2.6) | 3.21 (2.71–3.4) | 3.74 (3.4–4.3) | [b]**0.0001** | [b]NS |
| | | | | [c]**0.0018** | [c]NS |
| **Clinical attachment level (CAL)** **Median (IQR)** | | | | ¥0.0001 | |
| Colombia | 0.40 (0.1–1.7) | 2.6 (2.5–3.86) | 4.19(3.3–4.8) | [a]**0.0001** | [a]NS |
| Spain | 0.3 (0.1–1.6) | 3.4 (3.1–3.6) | 4.29 (3.8–5.3) | [b]**0.0001** | [b]NS |
| | | | | [c]**0.0004** | [c]NS |

n = number of patients; SD = standard deviation; IQR = interquartile range; NS = No statistically significant differences (p>0.05); Statistically significant differences (p<0.05)

¥ = differences between groups including the entire population (stages and health/gingivitis group)

**a** = differences between health/gingivitis and stage I-II periodontitis

**b** = differences between health/gingivitis and stages III-IV periodontitis

**c** = differences between stages I-II and stages III-IV. Periodontitis. Differences between countries compare each stage between countries.

However, when comparing periodontitis samples according to stage, *Peptostreptococcaceae* [XI] [G-9] *[Eubacterium] brachy* and *Desulfobulbus* sp. HMT 041 expression was higher in stage III-IV samples (p ≤ 0.05). However, *P. gingivalis* was only associated with stage I–II periodontitis when the entire sample was analyzed (Table 2).

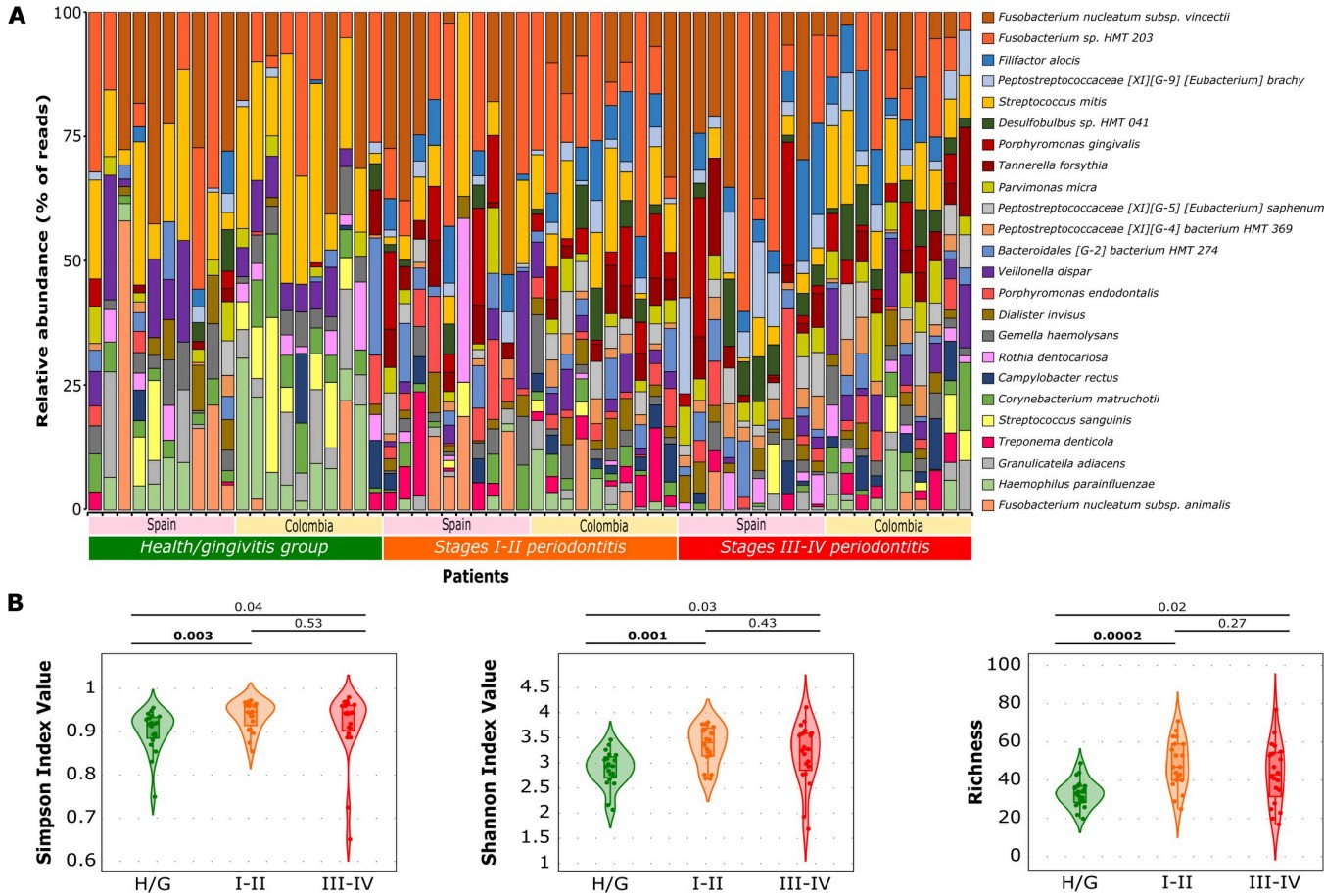

**Fig 1. Alpha diversity across evaluated samples by stages of periodontitis. (A)** The relative abundance of the 25 most abundant species across all samples did show. Samples are ordered accordingly to the stages: Periodontal health/gingivitis (Green), stages I-II periodontitis (Orange), and stages III-IV periodontitis (Red), and by geographic region: Spain (Pink box) and Colombia (Yellow box). **(B)** The comparison of alpha diversity metrics; Simpson's index, Shannon's index, and species richness across the different groups of samples (periodontal health/gingivitis [Green], stages I-II periodontitis [Orange], and stages III-IV periodontitis [Red]) did represent. Toplines show the p-values of Kruskal-Wallis-U Mann Whitney tests for the comparisons among groups.

## Comparison of subgingival microbiome composition between Spain and Colombia at different stages and grades of periodontitis

In an analysis by country, there were no statistically significant differences in the alpha diversity and richness between individuals with periodontal health/gingivitis and both stages of periodontitis in samples from Spain ($p \geq 0.01$). However, when assessing only the samples from Colombia, there were statistically significant differences ($p \leq 0.01$) between periodontal health/gingivitis and periodontitis, irrespective of staging (Fig 4A and 4B).

When comparing health/gingivitis and stage I-II periodontitis samples in patients from Spain, the highest proportions of periodontitis species were *Porphyromonas endodontalis* and *T. forsythia* (Fig 5A). In Colombian samples, the highest proportions were observed for *F. alocis*, *P. gingivalis*, *Peptostreptococcaceae* [XI] [G-5] [*Eubacterium*] *saphenum*, *Treponema denticola*, *Dialister invisus*, *Lachnospiraceae* [G-8] *bacterium* HMT 500, *Peptostreptococcaceae* [XI] [G-4] *bacterium* HMT 369, *Peptoniphilaceae* [G-1] *bacterium* HMT 113, *Selenomonas sputigena*, and *Peptostreptococcaceae* [XI] [G-9] [*Eubacterium*] *brachy*, and a higher proportion of taxa without assignment. Furthermore, Colombian samples demonstrated greater diversity ($p<0.05$) (Fig 5B).

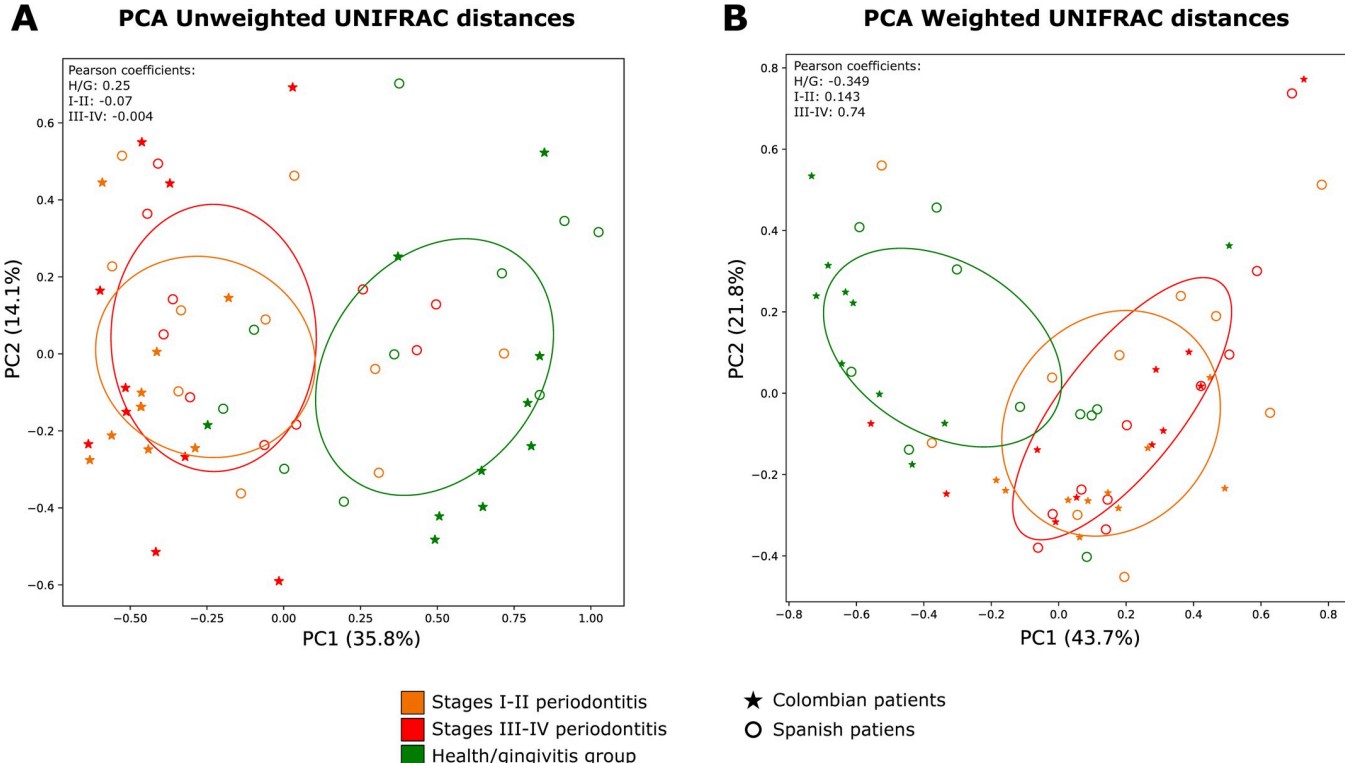

**Fig 2.** Principal components analysis of (A) unweighted and (B) weighted UNIFRAC distances across all samples. Each point represents a sample from the periodontal health/gingivitis group (Green), stages I-II periodontitis (Orange), and stages III-IV periodontitis (Red). Filled stars represent the country of origin of the samples for Colombia and empty circles for Spain.

Samples from both countries with stage III-IV periodontitis showed higher proportions of *F. alocis* and *Desulfobulbus* sp. HMT 041, *Peptostreptococcaceae* [XI] [G-9] [*Eubacterium*] brachy, *Peptostreptococcaceae* [XI] [G-5] [*Eubacterium*] saphenum and *Peptostreptococcaceae* [XI] [G-4] *bacterium* HMT 369. However, *Mogibacterium thymidum*, *Bacteroidales* [G-2] *bacterium* HMT 274, and *T. forsythia* were also observed in higher proportions in stage III-IV periodontitis samples from Spain (p<0.05) (Fig 6A). In comparison, Colombian samples showed statistically higher proportions of *P. gingivalis*, *T. denticola*, *Lachnospiraceae* [G-8] *bacterium* HMT 500, *Fretibacterium* sp. HMT 361, *Dialister invisus*, *Peptostreptococcaceae* [XI] [G-6] [*Eubacterium*] *nodatum*, and unassigned taxa (p<0.05) (Fig 6B).

Comparing stage I-II periodontitis with periodontal health/gingivitis samples, the highest species were unassigned taxa, *Streptococcus intermedius*, and *Haemophilus parainfluenzae* in samples from Spain. In samples from Colombia, the most abundant taxa were *S. mitis*, *H. parainfluenzae*, *Streptococcus sanguinis*, *Granulicatella adiacens*, *Rothia dentocariosa*, *Corynebacterium matruchotii*, *Leptotrichia hongkongensis*, *Rothia aeria*, and *Bergeyella* sp. (Fig 5A and 5B). Compared to periodontitis samples, the highest proportions of microorganisms associated with the periodontal health/gingivitis samples vs. stage III-IV periodontitis were *C. matruchotii* and unassigned taxa in both countries. In Spain, the level of *H. parainfluenzae* and *Escherichia coli* was found to be high (Fig 6A), whereas in Colombia, the level of *S. mitis*, *S. sanguinis*, *G. adiacens*, *L. hongkongensis*, and *Bergeyella* sp. was high (Fig 6B).

Only a few species had differences in their proportions among the distinct stages of periodontitis in Spain, such as *Peptostreptococcaceae* [XI] [G-9] [*Eubacterium*] brachy, *Desulfobulbus* sp. HMT 041 and *Peptostreptococcaceae* [XI] [G-4] *bacterium* HMT 369 (Fig 7A). In

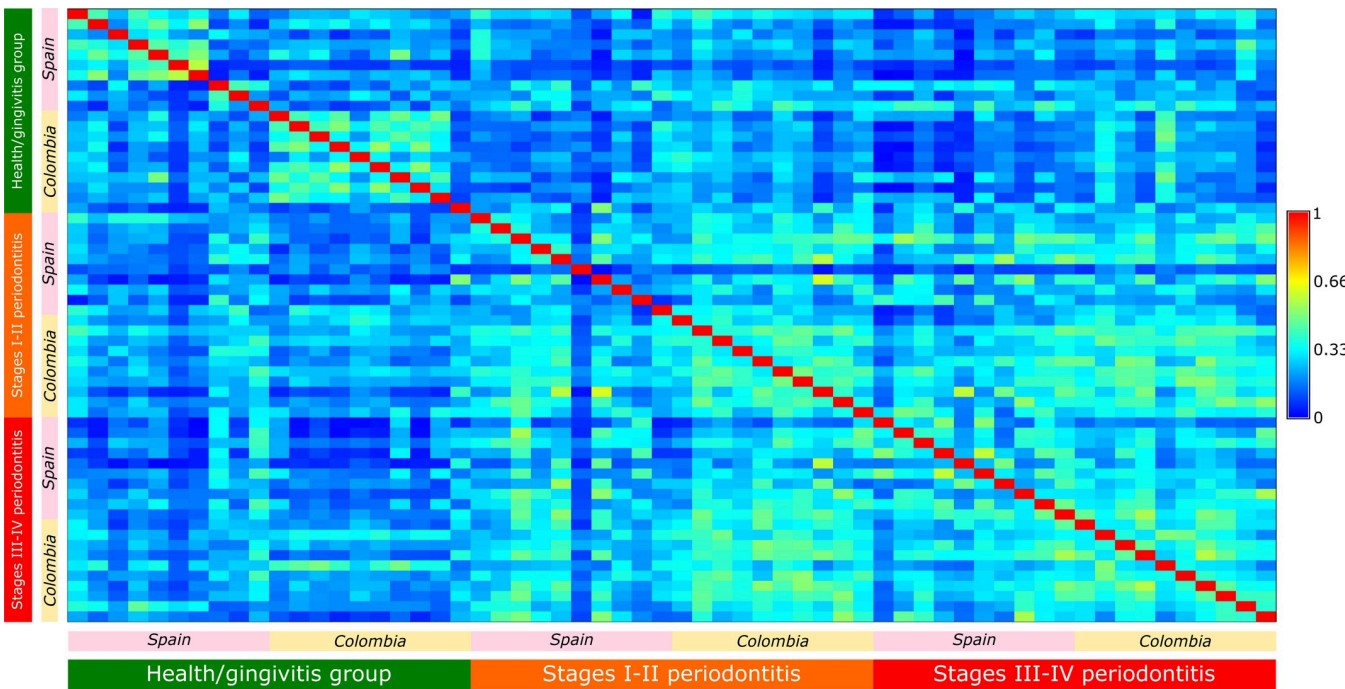

**Fig 3. Heatmap of the Bray-Curtis similarity index across all samples.** The samples are organized accordingly to country and levels of disease. The color blue represents maximum dissimilarity, while red shows maximum similarity (diagonal). Samples are ordered accordingly to the stage of the disease: Periodontal health/gingivitis (Green), stages I-II periodontitis (Orange), and stages III-IV periodontitis (Red), and by geographic region: Spain (Pink box) and Colombia (Yellow box).

Colombia, the different microorganisms did not show differences in proportions when samples were compared according the stage (Fig 7B). However, there were no significant differences in the studied groups between smokers and non-smokers in the periodontal health/gingivitis group and stages of periodontitis in these groups (data not shown).

When periodontitis samples were compared according to grade, *Peptostreptococcaceae* [XI] [G-9] *[Eubacterium] brachy*, *F. alocis*, and *Desulfobulbus* sp, HMT 041 discriminated between grades A/B (n = 18) and C (n = 22) but differed between countries. (Fig 8). However, microorganisms from the phylum *Bacteroidetes* such as *P. gingivalis* and *T. forsythia* did not differ between states or grades of periodontitis.

## Discussion

In this observational study of microbial samples collected from individuals with different periodontal conditions (periodontal health/gingivitis, stage I-II periodontitis, and stage III-IV periodontitis), the highest differences in the proportions of bacteria encountered in the subgingival microbiome in periodontitis, versus periodontal health/gingivitis samples, were in the new non-culturable or difficult to culture microbial genera. Only a few microbial genera showed differences when comparing periodontitis samples by staging. When comparing samples by country of origin, there were significant differences in the richness, diversity, and microbial composition of subgingival biofilm samples, even at similar stages.

Other studies have assessed the subgingival microbiome by comparing periodontal health/gingivitis samples with periodontitis samples using NGS technologies, reporting contradictory results [13–15,24]. These differences could be explained by different sequencing methods, inaccurate grouping systems among the obtained sequences, and inaccurate diagnostic criteria. However, to reduce bias due to erroneous classification in taxonomic groups, the analyses

**Table 2. Microbial diversity differences at species level between periodontitis groups.**

| Taxonomic ASV group | Health/gingivitis vs Stages I-II periodontitis | | Health/gingivitis vs Stages III-IV periodontitis | | Stages I-II vs Stages III-IV periodontitis | |
|---|---|---|---|---|---|---|
| | Difference in the mean of proportions* | p-value | Difference in the mean of proportions* | p-value | Difference in the mean of proportions** | p-value |
| *Bulleidia extructa* | | ns | -0.581 | **0.009** | | ns |
| *Campylobacter gracilis* | | ns | 0.848 | **0.009** | 1.057 | **0.01** |
| *Corynebacterium matruchotii* | 1.259 | 0.029 | 1.764 | **0.002** | | ns |
| *Desulfobulbus* sp. HMT 041 | | ns | -2.559 | **0.001** | -1.749 | 0.02 |
| *Filifactor alocis* | -2.617 | **0.0004** | -4.260 | **< 0.001** | | ns |
| *Fretibacterium fastidiosum* | -0.417 | **0.003** | -0.669 | **0.006** | | ns |
| *Lachnospiraceae* [G-8] *bacterium* HMT 500 | -0.869 | **0.005** | -0.981 | **0.002** | | ns |
| *Mogibacterium timidum* | -0.433 | 0.013 | -0.516 | **0.001** | | ns |
| *Peptostreptococcaceae* [XI][G-4] *bacterium* HMT 369 | -0.805 | **0.002** | -2.145 | **< 0.001** | -1.340 | **0.002** |
| *Peptostreptococcaceae* [XI][G-5] *[Eubacterium] saphenum* | -1.495 | **0.002** | -2.158 | **0.001** | | ns |
| *Peptostreptococcaceae* [XI][G-6] *[Eubacterium] nodatum* | -0.386 | 0.031 | -0.381 | **0.005** | | ns |
| *Peptostreptococcaceae* [XI][G-9] *[Eubacterium] brachy* | -0.723 | 0.020 | -3.421 | **0.002** | -2.698 | **0.01** |
| *Peptostreptococcus stomatis* | -0.821 | **0.006** | -0.908 | **0.002** | | ns |
| *Porphyromonas endodontalis* | -1.720 | **0.007** | -1.821 | 0.025 | | ns |
| *Porphyromonas gingivalis* | -3.333 | **0.002** | -3.449 | 0.014 | | ns |
| *Streptococcus intermedius* | 1.111 | **0.004** | 0.966 | 0.016 | | ns |
| *Tannerella forsythia* | -1.884 | **0.001** | -2.169 | **0.006** | | ns |
| *Treponema denticola* | -1.769 | **0.005** | -1.087 | 0.018 | | ns |
| *Treponema socranskii* | -0.418 | **0.009** | | | | |

Statistically significant differences (p ≤ 0.01) are in **bold**; ns = not statistically significant.

*Positive values indicate a higher mean proportion in periodontal health/gingivitis samples, while negative values show a higher mean proportion in samples with periodontitis.

** Positive values indicate a higher mean proportion in stages I-II periodontitis samples, while negative values show a higher mean proportion in stages III-IV periodontitis samples.

used ASVs instead of OTUs [12]. Clustering methods based on ASVs allow for error control because they differentiate between variants up to a single nucleotide, thus providing a higher resolution in the assignment of taxonomic categories compared to OTU analyses. However, there are still limitations in the representation of some organisms [25].

This study showed that the phylum *Bacteroidetes* was more abundant in periodontitis, irrespective of the stage or country, compared to health/gingivitis samples. Species from the phylum *Bacteroidetes* grouped by their genetic homology have previously been associated with the presence and progression of periodontitis in microbiological studies using both culture-based and molecular-based diagnostic methods [26,27]. In addition, the presence of a relatively high abundance of *Fusobacterium* species in both populations, particularly in periodontitis samples, suggests a relevant pathological role for these species, possibly as a bridge species in microbial succession between primary and later colonizers, thus allowing for more pathogenic microorganisms colonizing subgingival biofilms [28]. Together with the described significant relative abundance of known periodontal pathobionts in periodontitis samples from both countries,

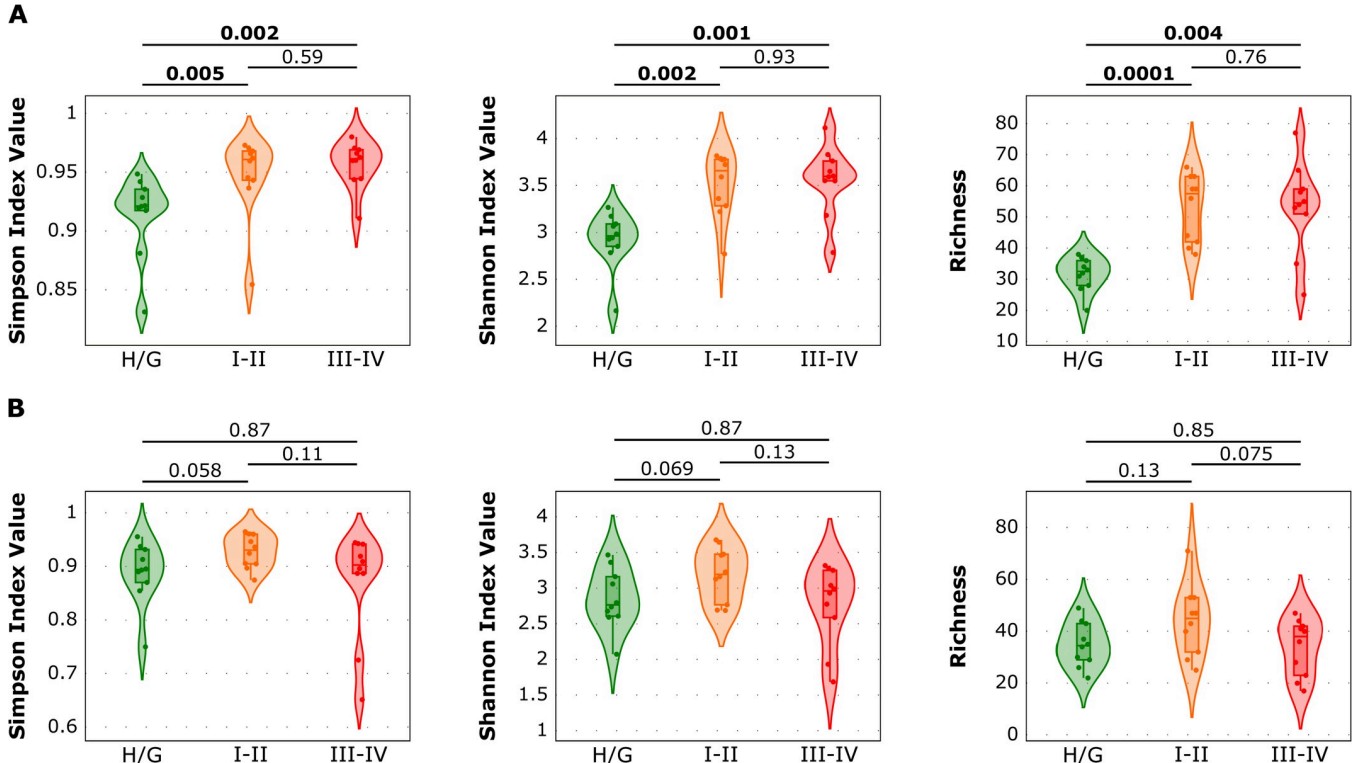

**Fig 4. Comparison of alpha diversity metrics (Simpson's index, Shannon's index, and species richness) by country.** Health/gingivitis [Green], stages I-II periodontitis [Orange], and stages III-IV periodontitis [Red]. (**A**) The samples from Colombia and (**B**) the samples from Spain are represented. Toplines show the p-values of the Kruskal-Wallis test for the comparisons among groups.

there was a relative abundance of new microorganisms (e.g., *F. alocis*), which may indicate that individuals with periodontitis may harbor a higher diversity and imbalance in the microbial ecology of the subgingival microbiome [28].

There is reason to believe that microbiota diversity is associated with health, and lower diversity is considered a marker of dysbiosis (microbial imbalance) in the gut [29]. However, this did not appear to occur in the subgingival microbiome. Although a greater alpha diversity of the oral microbiome was associated with periodontal health [30] or gingivitis [28,31] compared with periodontitis, other studies have reported greater diversity in periodontitis [14,32] or have not found differences [33]. In the present study, alpha diversity was higher in stage I-II and III-IV periodontitis than in periodontal health/gingivitis when the entire population was analyzed. However, these differences were due to the greater diversity present in the samples from Colombia. It is possible that the small sample size of this study, along with the pooled analysis of samples, may have underestimated the differences in the diversity of the samples. This diversity difference could also be explained by specific ecological pressures that may change microbial succession patterns and not always result in differences in the relative abundances of the microbiome [32]. In addition, other environmental factors, such as food consumption, socioeconomic factors, and access to dental care, may also influence it [29,34]. Another factor that could affect diversity is the status of clinical inflammation. However, in Spanish samples, the microbiome showed less diversity but more BoP in individuals with the same stage than in Colombian samples. Abuselme et al. [32] did not find bleeding to be associated with different alpha diversity or a distinct microbiome than sites without bleeding in periodontitis lesions. However, bleeding sites showed a higher total bacterial load. A higher

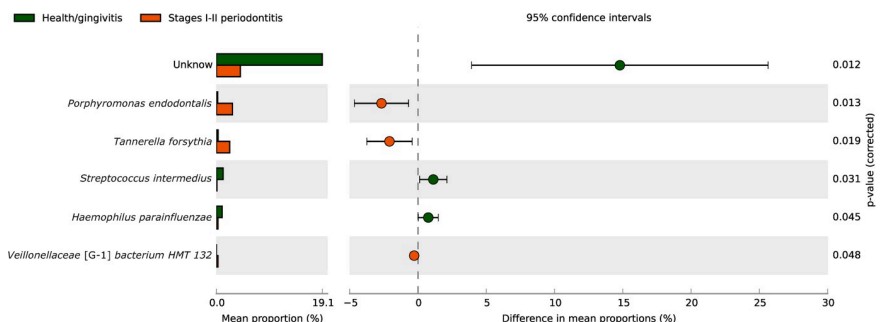

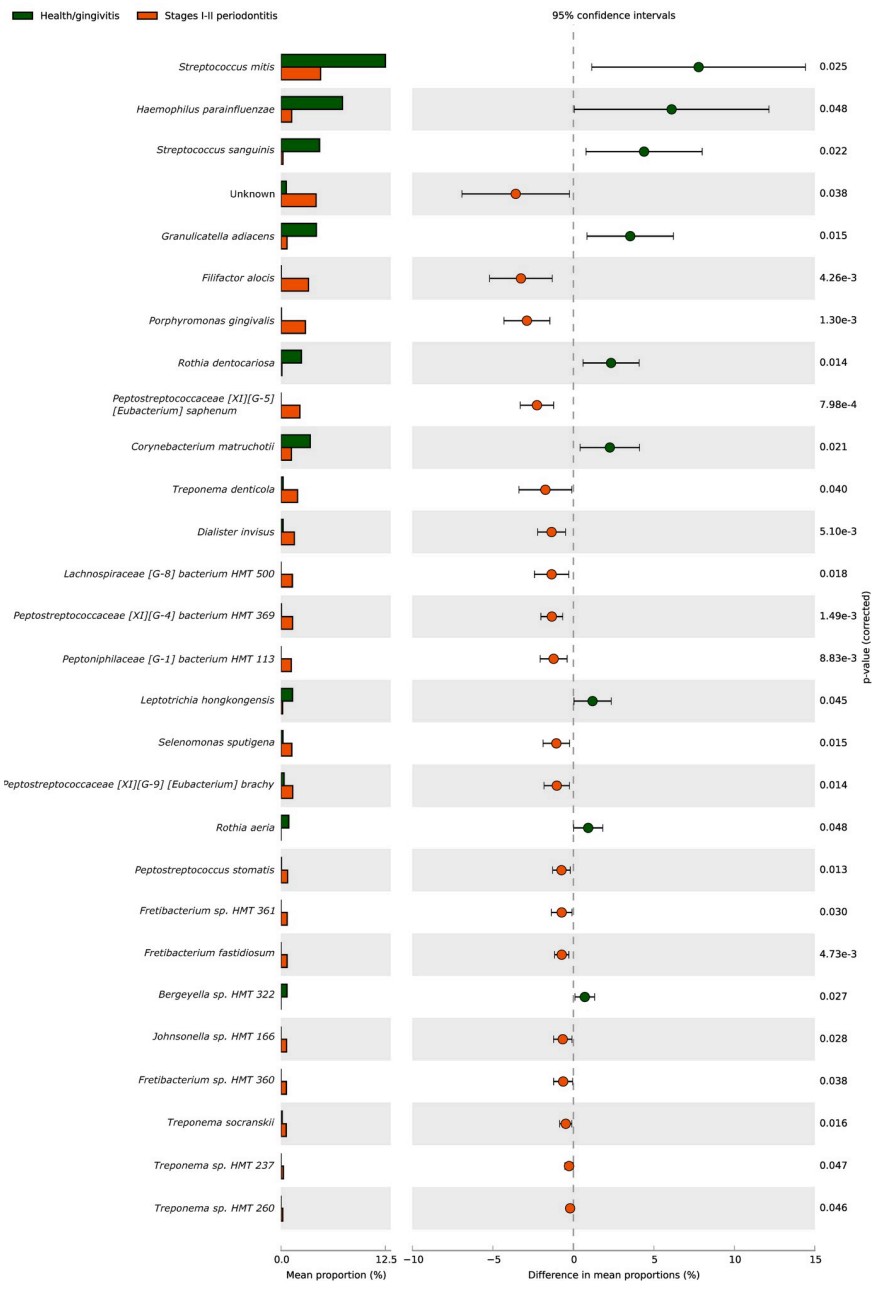

**Fig 5. A**. Species composition comparisons between health/gingivitis and stages I-II periodontitis in Spain **B**. Species composition comparisons between periodontal health/gingivitis and stages I-II periodontitis in Colombia.

diversity alone cannot be used as a predictor of health in the subgingival microbiome because it ignores the functions of specific species and how these may dictate the nature of bacterial interactions between them [35].

When comparing periodontitis versus periodontal health/gingivitis samples, irrespective of the stage or grade and the country, microorganisms that were difficult to culture or unculturable were highly associated with periodontitis. The results are consistent with similar association studies using NGS in populations from North America [14,36–38], Latin-American [39,40], Europe [13,41], and Asia [15,42,43].

The most significant outcome was the association between *F. alocis* and *Desulfobulbus* sp. HMT 041 *Peptostreptococcaceae* [XI][G-9] *[Eubacterium] brachy*, *Peptostreptococcaceae* [XI] [G-4] *bacterium* HMT 369, and *Peptostreptococcaceae* [XI][G-5] *Eubacterium saphenum* with stage III-IV periodontitis observed in both countries. These non-traditional periodontal pathogens have been associated with periodontitis in other studies in different populations [31,37,40]. Although these microorganisms can differentiate between health/gingivitis and different stages of periodontitis, there is not always a difference between the stages of periodontitis. Only a few species were differentially associated with periodontitis according to staging: *Peptostreptococcaceae* [XI][G-9] *[Eubacterium] brachy*, *Desulfobulbus* sp. HMT 04, *Peptostreptococcaceae* [XI][G-4] *bacterium* HMT 369 differentiated the stages of periodontitis in Spain but not in Colombia. A recent microbiological culture study found no differences in periodontopathic microorganisms between stages [44]. Although some species can differentiate between the stages, these species only increased in proportion. Only *Desulfobulbus* sp. HMT 04 can be identified in stage III-IV periodontitis and could be used to differentiate between stages and grades.

*P. gingivalis* was only associated with stage III-IV periodontitis in samples from Colombia, whereas *T. forsythia* was associated with stage III-IV periodontitis in samples from Spain. Other NGS results did not show an association between *P. gingivalis* and periodontitis lesions [14,31,41,43]. Although the participants from Spain were slightly more likely to be smokers, no differences were found between smokers and non-smokers in terms of the microbiome. In another study, cigarette smoking considerably affected the subgingival bacterial ecology of patients with periodontitis [45]. A small number of smokers participated in our study, which may have contributed to the loss of these differences.

*F. alocis* is an asaccharolytic anaerobic fastidious gram-positive rod associated with periodontitis because of its ability to survive in the periodontal pocket's oxidative stress-rich environment and alter microbial community dynamics by interacting with several oral bacteria [46]. Furthermore, this bacterium has shown relevant virulence capacities, such as resistance to oxidative stress, production of unique proteases and collagenases, and the ability to dysregulate the host response. In addition, *F. alocis* may alter bone metabolism via TLR2 activation [47]. In recent studies, *F. alocis* has been proposed to be included within the red complex of periodontal pathogens [48], probably due to these properties that enable this bacterium to colonize, survive, and outcompete other bacteria within the inflammatory environment of the periodontal pocket [48].

*Desulfobulbus* sp., previously uncultured, is the first human-associated representative of its genus. This sulfate-reducing deltaproteobacterium has adapted to the human oral subgingival niche by losing its biosynthetic abilities and reducing metabolic independence, environmental sensing, and signaling capabilities. Similar to other known oral pathobionts, *Desulfobulbus* sp.

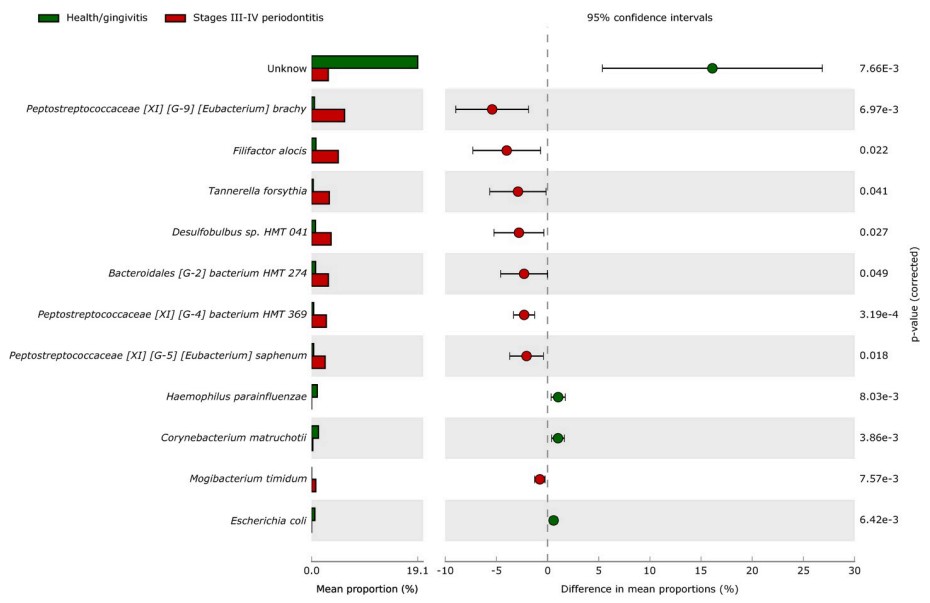

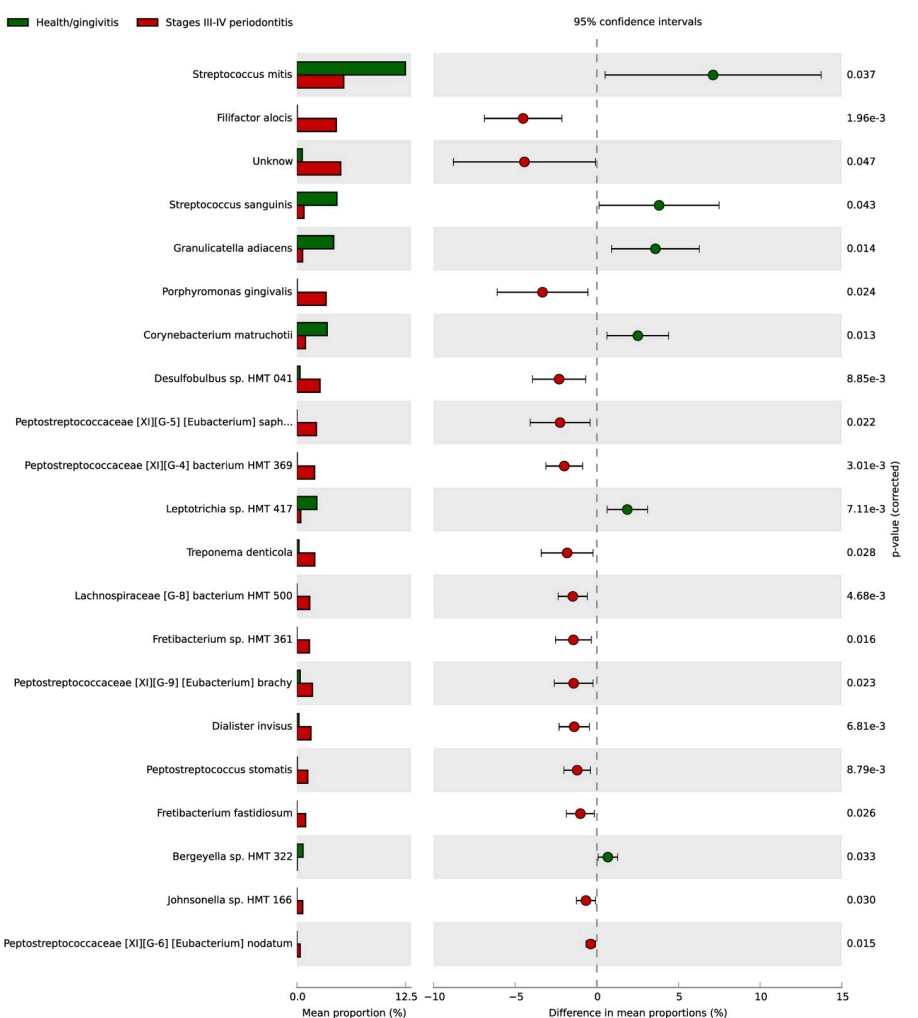

**Fig 6. A.** Species composition comparisons between health/gingivitis and stages III-IV periodontitis in Spain **B.** Species composition comparisons between health/gingivitis and stages III-IV periodontitis in Colombia.

can trigger a proinflammatory response in oral epithelial cells [49]. Its presence has been associated with pocket depth in patients with periodontitis compared to healthy sulci, suggesting a possible pathogenic role in periodontitis that should be further investigated [50].

Other microorganisms found in severe periodontitis were of the genus *Eubacterium*, which currently includes a heterogeneous group of gram-positive, non-spore-forming, anaerobic rods, many of which are slow-growing, fastidious, and are generally difficult to cultivate and identify [51]. Of this group, *E. saphenum* and *E. brachy* are obligate anaerobes that are moderate producers of acetate and butyrate as end products of human periodontal pockets [52], which have been identified as important virulence factors in the disease and have allowed them to be recognized as pathobionts associated with periodontal disease [53–55].

In summary, this study has shown that new unculturable or difficult to culture microorganisms were associated with periodontitis within the subgingival microbiome, but their relative proportions differed when the samples were compared by country. *F. alocis* and *Desulfobulbus* sp. HMT 041 was found in higher proportions when comparing stage III-IV periodontitis with periodontal health/gingivitis samples in both countries, whereas *T. forsythia* showed higher proportions in Spain and *P. gingivalis* in Colombia. These new putative periodontal pathogens should be investigated further in prospective studies.

This study had several strengths and limitations. This study used a small sample size, similar to most metagenomic studies. However, the results showed significant differences, with high reliability (p<0.01) in the entire sample and acceptable reliability when the samples were compared by country (p<0.05). The results of this study cannot be extrapolated to the entire population. Studies with large sample sizes should be conducted to confirm these findings using NGS or qPCR techniques.

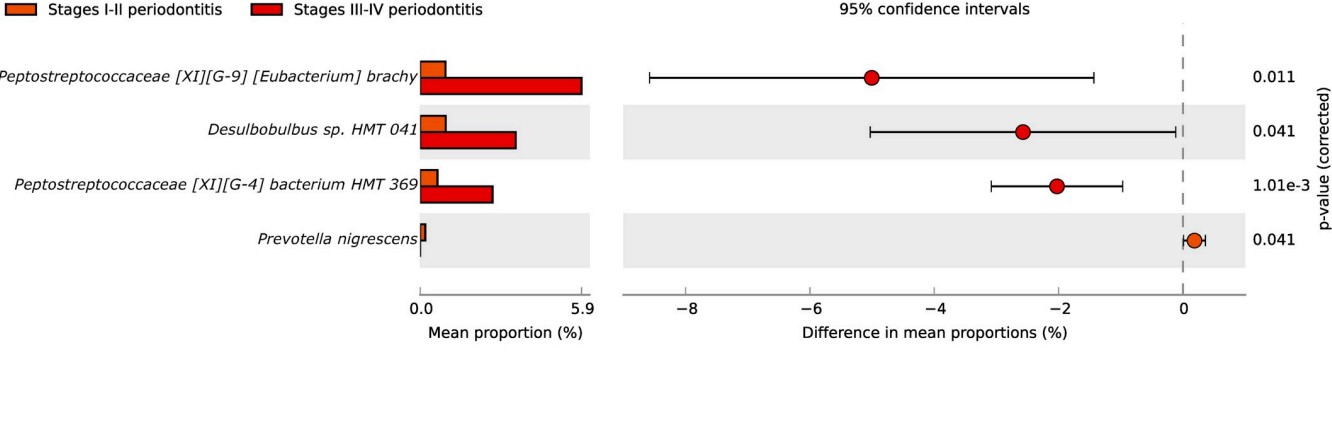

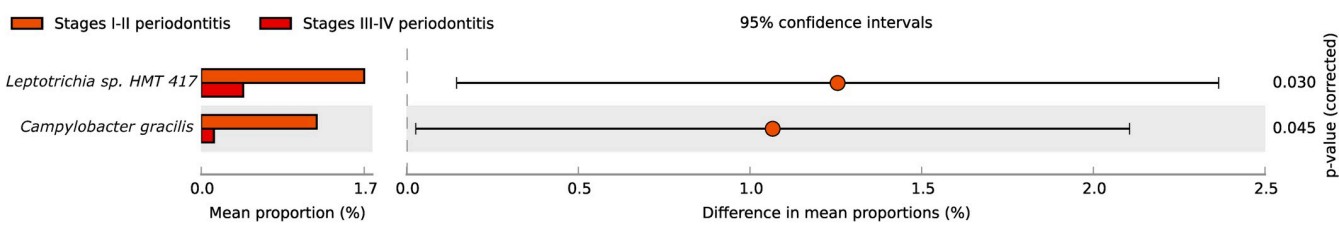

**Fig 7. A.** Species composition comparisons between stages I-II periodontitis and stages III-IV periodontitis in Spain **B.** Species composition comparisons between stages I-II periodontitis and stages III-IV periodontitis in Colombia.

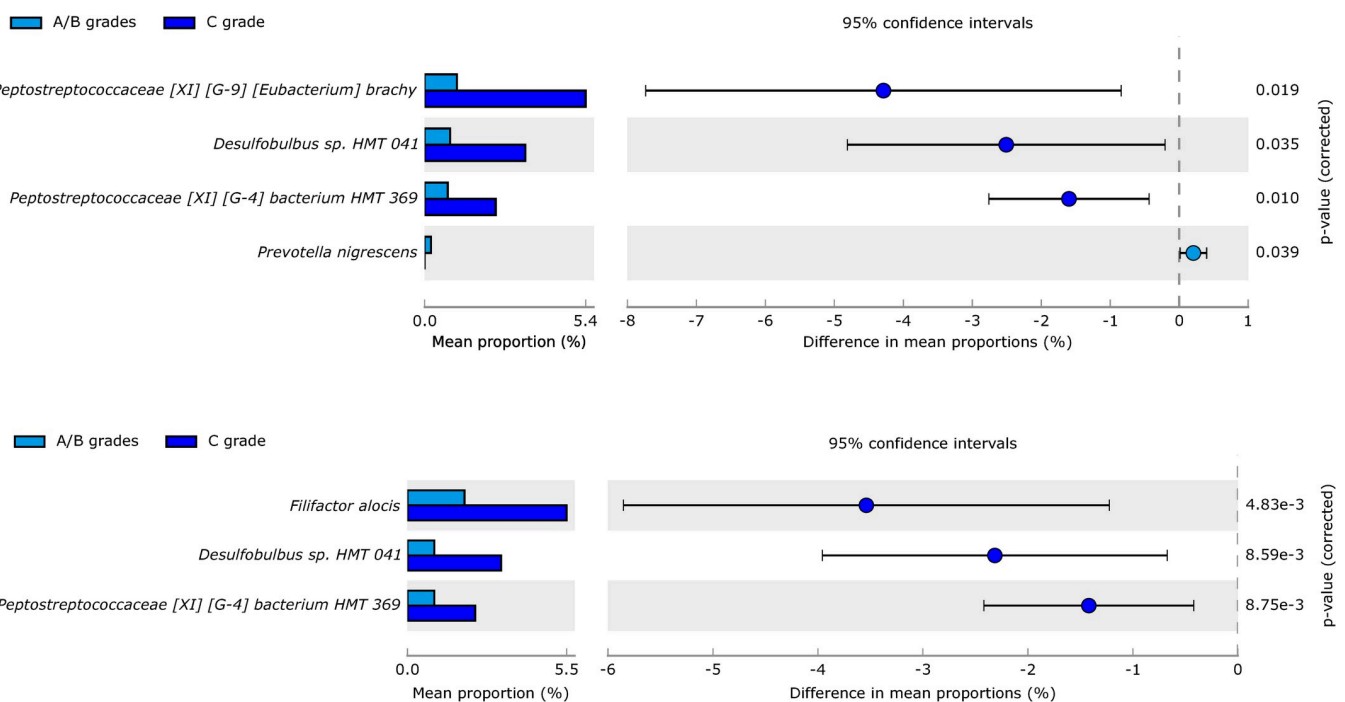

**Fig 8. A.** Species comparisons between periodontitis grade A/B and periodontitis grade C in Spain with statistical significance **B.** Species comparisons between periodontitis grade A/B and periodontitis grade C in Colombia.

## Strength and limitation

This study uses a tiny sample like most metagenomic studies. However, the results had found significant differences with high reliability (p<0.01) in the entire sample and acceptable reliability when the samples were compared in each country (p<0.05). The results of this study cannot be extrapolated to the entire population and require the evaluation of large sample sizes to confirm these findings using NGS or qPCR techniques in the future. However, describe a core microbiome of uncultivable microorganisms in periodontitis presented in both countries.

## Acknowledgments

We are grateful to the Schools of Dentistry and to the microbiological laboratories of both, the University Complutense of Madrid (UCM) and El Bosque University, and especially for the Molecular Genetics and Antimicrobial Resistance Unit, International Center for Microbial Genomics of Universidad El Bosque, Bogotá, Colombia.

## Author Contributions

**Conceptualization:** Gloria Inés Lafaurie, Roquelina Pianeta, Diana Marcela Castillo, David Herrera, Lorena Diaz, Mariano Sanz.

**Data curation:** Roquelina Pianeta, Diana Marcela Castillo, Yormaris Castillo.

**Formal analysis:** Gloria Inés Lafaurie, Yineth Neuta, Rafael Ríos, Mauricio Pacheco-Montealegre.

**Supervision:** Gloria Inés Lafaurie, David Herrera, Jinnethe Reyes, Lorena Diaz.

**Writing – original draft:** Gloria Inés Lafaurie, Yineth Neuta, Rafael Ríos, Roquelina Pianeta, Jinnethe Reyes, Lorena Diaz.

**Writing – review & editing:** Gloria Inés Lafaurie, Yineth Neuta, Rafael Ríos, Roquelina Pianeta, Diana Marcela Castillo, David Herrera, Yormaris Castillo, Mariano Sanz, Margarita Iniesta.

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
