## [Decision Letter · Decision Letter 0]

2 Aug 2022

PONE-D-22-12902Differences in the Subgingival Microbiome According to Stage of Periodontitis: A Comparison of Two Geographic RegionsPLOS ONE

Dear Dr. Lafaurie,

Thank you for submitting your manuscript to PLOS ONE. After careful consideration, we feel that it has merit but does not fully meet PLOS ONE’s publication criteria as it currently stands. Therefore, we invite you to submit a revised version of the manuscript that addresses the points raised during the review process.

We look forward to receiving your revised manuscript.

Kind regards,

Hansel McClear Fletcher, Ph.D.

Academic Editor

PLOS ONE

Journal Requirements:

2. Please specify how the sample size was determined and if any sample size calculation was conducted.

Please provide additional details regarding participant consent. In the ethics statement in the Methods and online submission information, please ensure that you have specified (1) whether consent was informed and (2) what type you obtained (for instance, written or verbal, and if verbal, how it was documented and witnessed). If your study included minors, state whether you obtained consent from parents or guardians. If the need for consent was waived by the ethics committee, please include this information.

Additional Editor Comments:

Please address the comments of the reviewers.

Reviewers' comments:

Reviewer's Responses to Questions

**Comments to the Author**

1. Is the manuscript technically sound, and do the data support the conclusions?

Reviewer #1: Yes

Reviewer #2: Yes

2. Has the statistical analysis been performed appropriately and rigorously? 

Reviewer #1: Yes

Reviewer #2: Yes

3. Have the authors made all data underlying the findings in their manuscript fully available?

Reviewer #1: Yes

Reviewer #2: Yes

4. Is the manuscript presented in an intelligible fashion and written in standard English?

Reviewer #1: Yes

Reviewer #2: Yes

5. Review Comments to the Author

Reviewer #1: To my view this is an excellent paper and demonstrates the lack of significant differences between oral bacterial populations in Spain and Colombia which is actually very interesting taking into account ethnic and nutritional differences. Moreover the paper is also a brilliant demonstration that research in Dentistry may be of high quality when performed in the shared fields with advanced sciences such as Microbiology. I reccommend publication.

Reviewer #2: In the manuscript entitled “Differences in the Subgingival microbiome according to the stage of periodontitis: A comparison of two geographic regions” submitted by Gloria Ines Lafaurie et al., authors have compared the differences of the subgingival microbiome in periodontal patients (stages I/II and III/IV compared to H/G) from two different geographical populations (Spain and Colombia). In recent years several studies have been done to study the subgingival microbiome from periodontal patients from a specific country/region but the results comparing two different geographic regions are limited.

In general, the results presented here will contribute to better understand the diversity and composition of the subgingival microbiota from periodontal patients. The sequencing results are well explained and the results are discussed in context with other similar studies with contradictory results. I do not have any major comments for this manuscript.

Minor comments

1. Line 88: different periodontal ‘diagnosis’ instead of ‘diagnoses’.

2. Line 154: Put a period after Nanodrop 2000).

3. Throughout in the manuscript, the names of some bacteria are written in scientifically wrong way. For example, Filifactor alocis is written as F. alocis in early manuscript and later again as Filifactor alocis. This error is repeated at several places with names of different bacteria. Please carefully review the manuscript for these errors and correct them.

4. Lines 440-449: References are mixed up (line 443, ref 46 is 47; line 446, ref 47 is 48; line 447, ref 48 is 47; line 449, ref 46 is 47).

5. Line 446: “In recent studies, F. alocis has been included within the red complex periodontal pathogens”. F. alocis has only been proposed in recent studies to be included within the red complex bacteria but never been referred as red complex bacteria. Therefore, this sentence should be written as “In recent studies, F. alocis has been proposed to be included within the red complex periodontal pathogens”.

6. References 47, 48 and 49: Name of bacteria should be italicized.

6. PLOS authors have the option to publish the peer review history of their article (what does this mean?). If published, this will include your full peer review and any attached files.

Reviewer #1: **Yes: **Miguel Viñas

Reviewer #2: No

---

## [Author Response · Author response to Decision Letter 0]

5 Aug 2022

Journal Requirements

1. Please ensure that your manuscript meets PLOS ONE's style requirements, including those for file naming

Answer: Changes have been made, so we believe that the manuscript meets now PLOS ONE's style requirements.

2. Please specify how the sample size was determined and if any sample size calculation was conducted.

Answer: Microbiome data usually depicts a high variability and diversity, not only relative to the counts and proportions of the predominant bacteria, but also in the identification of phylotypes and unculturable bacteria, which is even more difficult to relate to previous reports. Furthermore, oral microbiome dynamics are relatively dependent on the local microenvironment and there are relevant differences even among different sites of the oral cavity. These circumstances make the sample size calculation to detect significant differences among different populations an insurmountable task. We chose a sample size similar to the other investigations in other geographical locations studying the subgingival microbiome in periodontitis. Our goal was to achieve a sample size that would identify significant differences according to the subject’s periodontal status using the new classification of periodontal diseases with a power of 80%, a significance level of 95% (p< 0.05), and a confidence interval of 99%.

Please provide additional details regarding participant consent.

Answer: This cross-sectional observational study was conducted once the respective clinical ethics committees approved the study protocol (approval 012-2018 in Colombia and 18/127-E in Spain). This study adhered to the international ethical guidelines of the Declaration of Helsinki for human experimentation. The participants were informed of the characteristics of this observational study and asked to sign an informed consent if fulfilling the entrance criteria and were willing to participate. Both the written information sheet and informed consent form had been previously approved by the respective ethics committees. This information on signing the informed consent is included in the article.

3. Please note that in order to use the direct billing option the corresponding author must be affiliated with the chosen institute.

Answer: the corresponding author is affiliated with the chosen institute

4. Please review your reference list to ensure that it is complete and correct.

Answer: The reference list was reviewed, following the suggestion.

Reviewers' comments

1. Line 88: different periodontal ‘diagnosis’ instead of ‘diagnoses’.

Answer: In line 88, the word ‘diagnoses’ has been changed to ‘diagnosis‘.

2. Line 154: Put a period after Nanodrop 2000).

Answer: In line 154, it was corrected.

3. Throughout the manuscript, the names of some bacteria are written in the scientifically wrong way. For example, Filifactor alocis is written as F. alocis in the early manuscript and later again as Filifactor alocis. This error is repeated at several places with names of different bacteria. Please carefully review the manuscript for these errors and correct them.

Answer: Bacteria names have been revised and corrected when needed.

4. Lines 440-449: References are mixed up (line 443, ref 46 is 47; line 446, ref 47 is 48; line 447, ref 48 is 47; line 449, ref 46 is 47).

Answer: In lines 440-449, the mixed references were corrected.

5. Line 446: “In recent studies, F. alocis has been included within the red complex periodontal pathogens”. F. alocis has only been proposed in recent studies to be included within the red complex bacteria but has never been referred to as red complex bacteria. Therefore, this sentence should be written as “In recent studies, F. alocis has been proposed to be included within the red complex periodontal pathogens”.

Answer: We agree and thank the reviewer´s suggestion, and we have made the suggested change in line 446.

6. References 47, 48, and 49: The name of bacteria should be italicized.

Answer: In references 47, 48, and 49, the names of bacteria are now written in italics, following the reviewer´s suggestion.

---

## [Editor Report · Decision Letter 1]

10 Aug 2022

Differences in the subgingival microbiome according to stage of periodontitis: A comparison of two geographic regions

PONE-D-22-12902R1

Dear Dr. Lafaurie,

We’re pleased to inform you that your manuscript has been judged scientifically suitable for publication and will be formally accepted for publication once it meets all outstanding technical requirements.

Kind regards,

Hansel McClear Fletcher, Ph.D.

Academic Editor

PLOS ONE

---

## [Editor Report · Acceptance letter]

15 Aug 2022

PONE-D-22-12902R1 

Differences in the subgingival microbiome according to stage of periodontitis: A comparison of two geographic regions 

Dear Dr. Lafaurie:

I'm pleased to inform you that your manuscript has been deemed suitable for publication in PLOS ONE. Congratulations! Your manuscript is now with our production department. 

Kind regards, 

on behalf of

Dr. Hansel McClear Fletcher 

Academic Editor

PLOS ONE